# Deep Reinforcement Learning for Modelling Protein Complexes

**Ziqi Gao** [*]
HKUST

**Tao Feng** [*]
HKUST (GZ)

**Jiaxuan You**
UIUC

**Chenyi Zi**
HKUST (GZ)

**Yan Zhou**
Createlink Technology

**Chen Zhang**
Createlink Technology

**Jia Li**[†]
HKUST (GZ)

## Abstract

AlphaFold can be used for both single-chain and multi-chain protein structure prediction, while the latter becomes extremely challenging as the number of chains increases. In this work, by taking each chain as a node and assembly actions as edges, we show that an acyclic undirected connected graph can be used to predict the structure of multi-chain protein complexes (a.k.a., protein complex modelling, PCM). However, there are still two challenges: 1) The huge combinatorial optimization space of $N^{N-2}$ ($N$ is the number of chains) for the PCM problem can easily lead to high computational cost. 2) The scales of protein complexes exhibit distribution shift due to variance in chain numbers, which calls for the generalization in modelling complexes of various scales. To address these challenges, we propose **GAPN**, a **G**enerative **A**dversarial **P**olicy **N**etwork powered by domain-specific rewards and adversarial loss through policy gradient for automatic PCM prediction. Specifically, GAPN learns to efficiently search through the immense assembly space and optimize the direct docking reward through policy gradient. Importantly, we design an adversarial reward function to enhance the receptive field of our model. In this way, GAPN will simultaneously focus on a specific batch of complexes and the global assembly rules learned from complexes with varied chain numbers. Empirically, we have achieved both significant accuracy (measured by RMSD and TM-Score) and efficiency improvements compared to leading PCM softwares. GAPN outperforms the state-of-the-art method (MoLPC) with up to 27% improvement in TM-Score, with a speed-up of $600\times$.

## 1 Introduction

Predicting the 3D structures of large protein complexes (Jumper et al., 2021; Hickman & Levy, 1988; Benesch et al., 2003) is crucial for understanding their biological functions and mechanisms of action. This knowledge is essential for studying various cellular processes, such as signal transduction (Ditzel et al., 1998), gene regulation (Will & Lührmann, 2011), and enzymatic reactions (Gao et al., 2006). The groundbreaking AlphaFold-Multimer (Evans et al., 2021) has presented great potential in directly predicting 3D complex structures from their amino acid sequences.

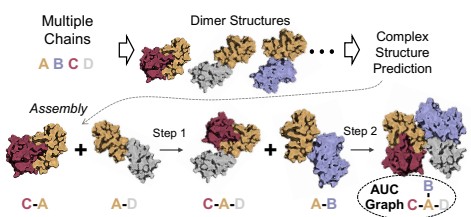

Figure 1: The assembly process applied for the PCM problem. Each incoming chain has the opportunity to dock onto any already docked chain.

However, it faces difficulties in maintaining high accuracy when dealing with complexes with a larger number ($> 9$) of chains (Bryant et al., 2022a; Burke et al., 2023; Bryant et al., 2022b).

Recently, a novel setting for protein complex modelling (PCM) has emerged and achieved significant success. These methods (Bryant et al., 2022b; Aderinwale et al., 2022; Esquivel-Rodríguez

---
[*]Equal contribution.

[†]Correspondence to: Jia Li (jialee@ust.hk).

et al., 2012; Burke et al., 2023) suggest that since numerous advanced protein-protein docking techniques (Ganea et al., 2021; Wang et al., 2023) are available, why not utilize them to sequentially assemble protein chains into complexes? They have demonstrated the ability to perform PCM on very large complexes (with up to 30 chains). However, the combinatorial optimization space (COS) of the PCM problem greatly limits the inference efficiency and effectiveness of these methods. As shown in Fig. 1, real-world data indicates that in a $N-$chain complex, the interaction of each chain may involve more than two chains. Therefore, the assembly can be determined by an acyclic, undirected, and connected (AUC) graph, which leads to the COS of $N^{N-2}$ (Shor, 1995).

Designing a proper deep learning (DL) framework holds great promise in handling the vast COS. However, directly learning from available complexes can be difficult since the data scarcity will increase as the chain number grows. For example, when the chain number exceeds 10, the average number of available complexes from the PDB database (Berman et al., 2000) is less than 100. Due to the inherent variability in the chain numbers of complexes, DL models are likely to face challenges in learning adequate knowledge for assembling large complexes (Aderinwale et al., 2022). Based on the above statements, we summarize two main challenges of the PCM problem as follows: **1) How to efficiently explore the vast combinatorial optimization space to find the optimal assembling action? 2) How to ensure that the DL model can effectively generalize to complex data with limited availability?**

In this paper, we propose GAPN, a generative adversarial policy network powered by domain-specific rewards and adversarial loss through policy gradient for automatic PCM prediction. Specifically, to address the first challenge, we design GAPN as a deep reinforcement learning (DRL) agent that operates within a complex assembling environment. The DRL agent can actively explore the vast combinatorial optimization space (COS) for assembling large complexes beyond samples through trial-and-error interaction with the environment (Zhang et al., 2019; Samsuden et al., 2019). Moreover, it incorporates a balance between exploration (trying new actions) and exploitation (sticking with known good actions) to ensure that the agent will not get stuck in local optima and continue to explore other potentially better solutions (Ladosz et al., 2022; Gupta et al., 2018a). The above two aspects make the GAPN agent able to efficiently explore the vast combinatorial optimization space to find the optimal assembling action. In response to the second challenge, we introduce an adversarial reward function to guide the learning process and enhance the receptive field of our model. The adversarial reward function incorporates the global assembly rules learned from complexes with varied chain numbers to enhance the generalization of GAPN. We empirically validate that the proposed GAPN model has high efficiency in exploring the vast COS in PCM problem, and can effectively generalize to complex data with limited availability. In summary, we make the following contributions:

- We propose GAPN, a fast, policy-based method for modeling large protein complexes, where GAPN functions as a DRL agent to efficiently explore correct assembly actions.

- We design an adversarial reward function that offers a global view of assembly knowledge, ensuring that GAPN can generalize over limited complex data.

- Quantitative results show that GAPN achieves superior structure prediction accuracy compared to advanced baseline methods, with a speed-up of $600\times$. Moreover, we contribute a dataset containing non-redundant complexes.

## 2 RELATED WORK

**Protein Complex Modelling.** The task of protein complex modelling (PCM) aims to predict the complex 3D structure given the information of all its chains. As shown in Table 1, existing PCM methods can be divided into two categories, namely one-time forming (OTF) and one-by-one assembling (OBOA). OTF-based methods can typically produce PCM results directly from the sequence or chain-level 3D structures. Specifically, the modified AlphaFold system, referred to as AlphaFold-Multimer (Evans et al., 2021), can handle up to 9-chain complexes, but it is very time-consuming. Moreover, Ji et al. (2023) proposed SyNDock, the learnable group synchronization approach for forming complexes in a globally consistent manner. SyNDock mainly claims significant speed-up, but the performance and the maximum chain number it can handle are both limited. Another PCM setting mainly involves two optimization aspects: the dimeric structures and the docking

path. Optimization approaches such as reinforcement learning (Aderinwale et al., 2022) and genetic algorithm (Esquivel-Rodríguez et al., 2012) are proposed for refining the dimeric structures. These methods are typically with low efficiency, and their max chain No. is also not satisfactory. MoLPC (Bryant et al., 2022b) assumes that the dimeric structures are given, thus only predicting the correct docking path. In this way, it is the first to demonstrate that the structures of complexes with up to 30 chains can be accurately predicted. However, its application of a tree search-based non-deep learning technique often leads to complete failure due to the lack of generalization.

| Methods | Assembly | Opt. Dimer | Opt. Path | Max Chain No. | Speed-Up |
|---|---|---|---|---|---|
| CombDock Inbar et al. (2005) | ✔ | ✔ | ✔ | 6 | 1.07 |
| Multi-LZerD Esquivel-Rodríguez et al. (2012) | ✔ | ✔ | ✔ | 6 | 1.33 |
| RL-MLZerD Aderinwale et al. (2022) | ✔ | ✔ | ✗ | 5 | 1 |
| ESMFold Lin et al. (2023) | ✗ | ✗ | ✗ | 4 | 213.65 |
| AlphaFold-Multimer Evans et al. (2021) | ✗ | ✗ | ✗ | 9 | 4.53 |
| MoLPC Bryant et al. (2022b) | ✔ | ✗ | ✔ | 30 | 3.02 |
| GAPN | ✔ | ✗ | ✔ | 60 | 1934.27 |

Table 1: Technical comparison between PCM methods. Assembly: PCM in the assembly fashion; Opt. dimer: Dimer structures will be optimized; Opt. Path: Docking path will be optimized; Max Chain No.: The maximum chain number that the method can handle. Speed-up: Efficiency speed-up compared to the RL-MLZerD baseline.

**Reinforcement Learning.** Over the past few years, RL has emerged as a compelling approach for various optimization tasks. The unique ability of RL agents to learn optimal policies from interaction with an environment enables them to collect and utilize more data, which enhances their ability to find excellent solutions to optimization problems with large search space. Several studies (Vinyals et al., 2015; Bello et al., 2016; Nazari et al., 2018) utilize RL and attention architecture to solve classic NP-hard problems with large and complex search space, such as TSP (Vinyals et al., 2015) and VRP (Bello et al., 2016; Nazari et al., 2018) problems. Khalil et al. (2017) and Fan et al. (2020) propose to introduce the graph architecture into the RL framework, so that RL can be used to solve large-scale graph optimization problems, such as finding the most influential nodes on the graph and optimizing the properties of the graph network. However, when applying RL to our problem, it still faces the generalization problem in scenarios with varied chain numbers. Therefore, in our paper, we design an adversarial reward function to guide the learning of the RL agent, which incorporates prior knowledge of multiple optimization scenarios.

**Adversarial Training** Adversarial training has been widely used to learn the underlying data distribution and derive a generator and a discriminator. The generator can generate synthetic data that is similar to the real data distribution, while the discriminator outputs the similarity between the input and real data. Adversarial training is widely used in many aspects, such as text generation (Zhang et al., 2017), imitation of human behavior (Gupta et al., 2018b), protein generation (Gupta & Zou, 2019). Adversarial training also enhances RL's ability to solve complex optimization problems by incorporating prior knowledge specified by a good oracle (Ho et al., 2016; Liu et al., 2018). Inspired by existing research, in this paper, we design an adversarial reward function to enhance the generalization ability of the RL agent based on the framework of adversarial training, which incorporates the global assembly rules learned from complexes with varied chain numbers.

## 3 PROPOSED APPROACH

### 3.1 PROBLEM DEFINITION

We are given a set of $N$ chains to form a protein complex. The $i-th$ chain consists of $n_i$ residues, which is characterized by its amino acid sequence $s^i$ and 3D structure in its undocked state $X_i \in \mathbb{R}^{3 \times n_i}$. We focus on the one-by-one assembly setting, where the dimeric 3D structures of all possible pairs of chains are provided.

We define the assembly process as follows. Formally, we have a set of assembly actions $\mathcal{A} = \{(A_k^1, A_k^2)\}_{1 \leq k \leq N-1}$, with each element representing a pair of chain indexes. Then, a transformation function set $\mathcal{T} = \{(R_k, t_k)\}_{1 \leq k \leq N-1}$ is obtained by $R_k \tilde{X}_{A_k^1} + t_k = X'_{A_k^1}$, where $R_k, t_k$

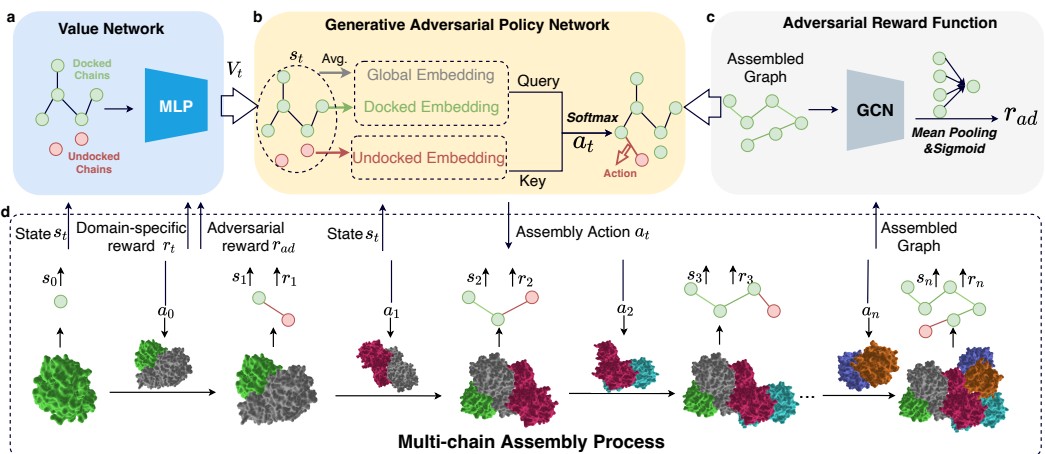

Figure 2: Overview of the proposed framework. It contains a Generative Adversarial Policy Network (GAPN) (b), which converts the input state $s_t$ into assembly action $a_t$ for the multi-chain assembly process (d). The GAPN is trained with the help of value network (a), which estimates the values come from two aspects: domain-specific rewards obtained through multi-chain assembly process (d) and adversarial rewards from adversarial reward function (c).

represent the rotation and translation transformation for the $k - th$ action, respectively. $\tilde{X}_{A_k^1}$ denotes structure of chain $A_k^1$ extracted from the dimeric structure of the pair of $(A_k^1, A_k^2)$ and $X'_{A_k^1}$ denotes the structure of the docked chain $A_k^1$. For the $k - th$ action, we dock the chain $A_k^2$ onto the docked chain $A_k^1$, such that $X'_{A_k^2} = R_k X_{A_k^2} + t_k$.

The task is to predict the assembly action set such that $\{RX'_i + t\}_{1 \leq i \leq N} = \{X_i^*\}_{1 \leq i \leq N}$, $\exists R \in SO(3), t \in \mathbb{R}^3$, by taking as inputs the amino acid sequences of all chains$\{s_i\}_{1 \leq i \leq N}$ of an input complex, where $X_i^*$ denotes the ground truth of the $i - th$ chain. In short, our goal is to predict the correct assembly action $\mathcal{A}$ such that the assembled complex structure matches the ground truth complex structure.

## 3.2 System Overview

To model the large protein complexes, we propose GAPN powered by domain-specific rewards and adversarial rewards through policy gradient. The structure overview is shown in Figure 2. The system mainly consists of four modules:

- **Generative Adversarial Policy Network.** It converts the input state to corresponding assembly action and interacts with multi-chain assembly process.

- **Adversarial Reward Function.** It incorporates the global assembly rules learned from complexes with varied chain numbers to enhance the generalization of GAPN, which converts the assembled graph into adversarial reward through Graph Convolutional Network (GCN).

- **Value Network.** It aims to estimate the value of the state and helps train the GAPN.

- **Multi-chain Assembly Process.** It describes the process of multi-chain assembly, state transfer, and assembly action effects, which has been discussed in Sec 3.1.

In the following sections, we will introduce these modules in detail.

## 3.3 Assembly of Protein Complexes as Markov Decision Process

We transform the assembly of protein complexes problem into a sequential decision-making problem, with the goal of predicting the correct assembly action that will bring the assembled complex structure to match the ground truth. Thus, we propose to formulate the problem using Markov Decision Process (MDP) (Puterman, 1990) in an RL setting, which has five parts:

- **Agent:** We consider the complex $\mathcal{M} = \{M_i\}_{1 \leq i \leq N}$ as the RL agent, who observes a set of $N$ chains and $n_i$ residues of each chain $M_i$.

- **State:** Since the agent is the complex $\mathcal{M}$, we define state at time interval $t$ as $s_t = (c_t, u_t, G_r)$, where $c_t$ denotes docked chain feature embeddings, $u_t$ denotes the undocked chain embeddings and $G_r$ denotes all chain embeddings in the complex. The chain embedding $e_i$ of $M_i$ is calculated through a protein pre-trained model, which will be introduced in Sec 3.4.

- **Action:** The action $a_t$ at time interval $t$ is to assemble two chains, which can be denoted by $(A_k^1, A_k^2)_{1 \leq k \leq N-1}$ with two elements representing a pair of chains.

- **Transition:** This component describes the process that the current state $s_t$ will transit to the next state $s_{t+1}$ after an action $a_t$ is taken. Specifically, in our state setting, the stack of all protein embeddings $G_r$ will remain unchanged. The stack of assembled protein feature embeddings $c_{t+1} = c_t \cup e_j$ and the stack of unassembled protein embeddings $u_{t+1} = u_t \backslash e_j$ will change due to newly docked chains, where $e_j$ denotes the embedding of the newly docked chain. With the implementation of assembly action $a_t$, the dimeric structure of chains will be updated, which is achieved with the transformation set $\mathcal{T}$ defined in Sec 3.1.

- **Reward:** This reward function $r$ is to measure how similar the assembled complex structure is compared with the real ones. Here, we define the rewards as a sum over domain-specific rewards $r_t$ and adversarial rewards $r_{ad}$. The domain-specific rewards are introduced to measure the similarity between the assembled complex structure and the corresponding ground truth. Specifically, we utilize RMSD to measure this similarity, which is inverted when used as a reward. Since the smaller the RMSD, the higher the similarity, we take negative RMSD values as domain-specific rewards. In order to enhance the generalization of the agent to limited data, we design an adversarial reward function incorporating the global assembly rules learned from complexes with varied chain numbers (see more discussion in Sec 3.5).

## 3.4 GENERATIVE ADVERSARIAL POLICY NETWORK

Having modeled the assembly of protein complexes as Markov Decision Process, we outline the architecture of GAPN, a policy network learned by the RL agent to act in the multi-chain assembly process. GAPN takes the state $s_t$ input and outputs the action $a_t$, which predicts an assembly action, as described in Section 3.3.

**Chain Embedding Calculation.** Through ESM (Lin et al., 2023) pre-training models, we obtain residue-level embeddings by inputting the entire sequence of each chain. ESM's pre-training captures the reflection of protein 3D structure within sequence patterns. To obtain the chain embedding $e$, we take the average of residue embeddings. It's worth noting that we refrain from fine-tuning the residue embeddings. This approach allows us to leverage the power of ESM's pre-training while maintaining the stability of the residue-level representations.

**Assembly Action Prediction.** The assembly action $a_t$ at time step $t$ is to decide which pair of proteins assembles together. The traditional RL approach is to perform one-hot encoding on all possible actions. However, these methods are difficult to apply to our problem: (1) In the protein assembly process, the number of possible protein pairs is constantly changing, which makes these methods hard to handle; (2) Since different protein complexes may have varied numbers of proteins, models trained based on these methods are difficult to generalize to different protein complexes. Inspired by the design of attention mechanism for supporting model's inputs with variable lengths (Vaswani et al., 2017), we first introduce a function $f_c$ to convert $c_t, G_r$ into dense vector $g_t$, and then regard the $g_t$ and $u_t$ as key and query to apply inner product for them and obtain the probability of deciding which pair of proteins assemble together,

$$g_t = f_c(c_t, G_r), \tag{1}$$

$$\pi(s_t) = \text{SOFTMAX}(f_a(g_t^T u_t)), a_t \sim \pi(s_t), \tag{2}$$

where $f_c$ and $f_a$ denote two MultiLayer Perceptrons (MLP) and $\pi$ denotes the GAPN.

## 3.5 Adversarial Reward Function

The vast combinatorial optimization space of large protein complexes makes it hard for RL to search for optimal assembly action sets to bring the assembled complex structure to match the ground truth. Moreover, the limited availability of these complex data further aggravates the difficulty of searching. In order to deal with this dilemma, the traditional approach is to mix complex data of different chains and utilize a unified RL policy network to learn (Beck et al., 2023). However, the huge differences in the features of complex data of different chains lead to poor training stability and performance (Nagabandi et al., 2018). Adversarial training can improve RL's ability to deal with complex problems by learning expert solutions as prior knowledge (Ho et al., 2016; Liu et al., 2018). Inspired by this, we utilize the correct assembly action sets of complex data in different chains as expert solutions, and employ the Generative Adversarial Network (GAN) framework (Goodfellow et al., 2020) to design the adversarial rewards $R(\pi_\theta, D_\phi)$ for RL learning:

$$\min_\theta \max_\phi R(\pi_\theta, D_\phi) = \mathbb{E}_{x \sim p_{\text{data}}(x)}[\log D_\phi(x)] + \mathbb{E}_{x \sim \pi_\theta}[\log(1 - D_\phi(x)))], \qquad (3)$$

where $\pi_\theta$ is the policy network parameterized by $\theta$, $D_\phi$ is the discriminator network parameterized by $\phi$, $x$ represents an assembled graph whose nodes represent chains and edges depict the connection relationship among the chains determined by the assembly action set. $p_{data}$ is the underlying data distribution of the ground truth assembly action sets of complex data with different chains. However, from equation 3, we can observe that only $D_\phi$ can be trained through stochastic gradient descent and $x$ as a graph is not differentiable in terms of parameters $\phi$. Therefore, we utilize $-R(\pi_\theta, D_\phi)$ as an auxiliary term with domain-specific rewards, which measures the similarity between the complex structure assembled from the generated assembly action set and the real ones.

The ability of Graph Neural Networks (Xu et al., 2018; Tang et al., 2022) to model the structured data enables it to perform well in many tasks (Gao et al., 2023a; Tang et al., 2023; Li et al., 2019b; Gao et al., 2023b). We employ the Graph Convolutional Networks (GCN) (Kipf & Welling, 2016; Li et al., 2019a; Cheng et al., 2023) to construct the discriminator network. Specifically, the nodes of the input complex graph are chain embeddings $E = e_i$, and the adjacency matrix $U$ of the input graph depicts the docked relationship among the chains determined by assembly action set. If the assembly action set contains $a_t = (A_K^1, A_k^2)$, then $U[A_K^1, A_k^2] = U[A_K^2, A_k^1] = 1$. We apply message passing with initial node embeddings $E = e_i$ and adjacency matrix $U$ to compute node embeddings for a total of $L$ layers:

$$H^{l+1} = \text{AGG}(\text{ReLU}(D^{-\frac{1}{2}} \tilde{U} D^{-\frac{1}{2}} H^l W^l)), \qquad (4)$$

where $H^l$ is the node embedding of the $l - th$ layer, $D$ is degree matrix of $U$, $\tilde{U} = U + I$ and $W^l$ is a trainable weight matrix of the $l - th$ layer. AGG$(\cdot)$ here is employed to denote an aggregation function that could be one of {MEAN, MAX, SUM, CONCAT} (Sun et al., 2023).

Based on the output $H^L$ of the last layer of GCN, the discriminator network applies an MLP layer, a SIGMOID function, and a MEAN Pooling function to get the final output $D_\phi(x)$:

$$D_\phi(x) = \text{SIGMOID}(\text{MEAN}(f_d(H^L))), \qquad (5)$$

where $f_d$ denotes the MLP layer and the MEAN Pooling function is to average embedding of the chain to obtain the representation of the entire graph.

## 3.6 Model Training

For the optimization of GAPN, we adopt Proximal Policy Optimization (PPO) (Schulman et al., 2017), which is one of the state-of-the-art policy gradient methods and has been widely used for optimizing policy networks. We show the objective function of PPO as follows:

$$\max \left( J_{\text{clip}}(\theta) \right) = \mathbb{E}_{\pi_{\theta_{\text{old}}}} \left[ \min \left( r_t(\theta) A^{\pi_{\theta_{\text{old}}}}(s, a), \text{clip}(r_t(\theta), 1 - \epsilon, 1 + \epsilon) A^{\pi_{\theta_{\text{old}}}}(s, a)) \right], \qquad (6)$$

$$r_t(\theta) = \frac{\pi_\theta(a|s)}{\pi_{\theta_{\text{old}}}(a|s)}, \qquad (7)$$

where $r_t(\theta)$ represents the ratio of the probability $\pi_\theta(a|s)$ of an action under the current policy to the probability $\pi_{\theta_{\text{old}}}(a|s)$ of the same action under the old policy and is clipped to the range of

---

**Algorithm 1** GAPN

---

**Input:** ground truth assembly graphs $G^r$.

1: Initialize $\pi_\theta$, $D_\phi$ with random weights for policy network and discriminator;
2: **for** $i$=0,1,2... **do**
3:     Generate a batch of assembly graphs $G_i^f \sim \pi_\theta$;
4:     **for** $k = 0, 1, 2...$ **do**
5:         Calculate a batch of rewards based on domain-specific reward function $r_t$ and adversarial function $r_{ad}$ for each state-action pair in sequences $G_i^f$;
6:         Sample generated assembly graphs $G_{ki}^f$ from $G_i^f$ with batch size $B$;
7:         Sample ground truth $G_k^r$ from $G^r$ with batch size $B$;
8:         Update $D_\phi$ based on equation (3) through stochastic gradient descent with the positive samples $G_k^r$ and negative samples $G_{ki}^f$.
9:     **end for**
10:    Update $\pi_\theta$ by maximizing equation (6) via the PPO method;
11: **end for**

---

$[1 - \epsilon, 1 + \epsilon]$ that makes the training of policy network more stable (Schulman et al., 2017), $A^{\pi_{\theta_{old}}}$ denotes the estimated advantage function which involves a learned value function $V(\cdot)$ to reduce the variance of estimation. Specially, $V(\cdot)$ is designed as an MLP that maps the state $s_t$ into a scalar.

We summarize details of the training process of GAPN in Algorithm 1. From the algorithm, we can first observe that a batch of generated assembly graphs and ground truth are sampled to train the discriminator (lines 6-8). The generated assembly graphs are regarded as negative samples and ground truths are regarded as positive samples to train discriminator via an Adam Optimizer (Reddi et al., 2019) (line 8). Then, we calculate a batch of rewards for the generated assembly graphs (line 5). Finally, we train the policy network by maximizing the expectation of reward via PPO algorithm (line 10).

## 4 EXPERIMENTS

### 4.1 EXPERIMENTAL SETUP

**Dataset.** Following MoLPC, we conduct experiments on protein complexes with the chain number $3 \leq N \leq 30$. We download all available complexes of their 'first biological assembly' version from the PDB database (Berman et al., 2000). Subsequently, we perform data filtering and sequence clustering using CD-HIT (Huang et al., 2010). We split the data based on the clusters to ensure that the sequence similarity between samples in the training and test sets will not exceed 50%. Finally, we obtained 7,063 PDBs for the training set and validation set, and 180 for the test set. A more detailed description of the data processing and filtering process can be found in Appendix B.

We apply two types of dimer structures, the ground-truth dimers (GT Dimer) and dimers predicted by AlphaFold-Multimer (AFM Dimer). For GT Dimer, we directly extract the dimer structures of all pairs of chains that have actual physical contacts. For those without physical contacts, we apply their dimers predicted with ESMFold (Lin et al., 2023) (due to its inference efficiency).

**Baselines.** We compare GAPN with recent advanced PCM methods. For non-assembly methods, we include (1) AlphaFold-Multimer (Evans et al., 2021) (AF-Multimer, an end-to-end deep learning model that takes only protein sequence as input for PCM); (2) ESMFold (Lin et al., 2023)(an end-to-end deep learning-based protein structure predictor without multiple sequence alignment). For assembly-based methods, we include (3) Multi-LZerD (Esquivel-Rodríguez et al., 2012) (a PCM predictor with genetic algorithm and a structure refinement procedure); (4) RL-MLZerD (Ader-inwale et al., 2022) (a reinforcement learning model for exploring potential dimer structures and docking path); (5) MoLPC (Bryant et al., 2022b) (a non-deep learning method with Monte Carlo tree search and plDDT information for guidance).

According to the maximum chain numbers that the baselines can handle, we set their available ranges of chain number Table 2 for testing (e.g., Multi-LZerD: $3 \leq N \leq 10$). We test all baselines

Table 2: **PCM prediction results.** The best performance for each metric is **bold** and the second best is underlined. '-' suggests that the method is not applicable to this range of chain number.

| Methods | Chain numbers $N$ | | | | | | | |
|---|---|---|---|---|---|---|---|---|
| | 3 | 4 | 5 | 6-10 | 11-15 | 16-20 | 21-25 | 26-30 |
| | TM-Score (mean) / RMSD (mean) | | | | | | | |
| Multi-LZerD | 0.54 / 19.17 | 0.49 / 21.58 | 0.40 / 33.05 | 0.24 / 56.73 | - | - | - | - |
| RL-MLZerD | 0.62 / 16.56 | 0.47 / 19.09 | 0.49 / 28.73 | 0.31 / 49.60 | - | - | - | - |
| AF-Multimer | 0.79 / 7.87 | 0.70 / 14.19 | 0.67 / 22.25 | 0.64 / 35.05 | 0.39 / 49.23 | - | - | - |
| ESMFold | 0.73 / 9.75 | 0.52 / 18.06 | 0.44 / 31.96 | 0.36 / 46.12 | - | - | - | - |
| MoLPC | 0.89 / 5.33 | 0.71 / 13.49 | 0.66 / 21.67 | 0.60 / 36.46 | 0.45 / 43.00 | 0.36 / 54.75 | 0.34 / 60.58 | 0.29 / 66.60 |
| **GAPN** | **0.95 / 1.16** | **0.81 / 8.52** | **0.79 / 16.96** | **0.65 / 32.37** | **0.62 / 37.58** | **0.48 / 50.05** | **0.38 / 55.76** | **0.36 / 59.13** |

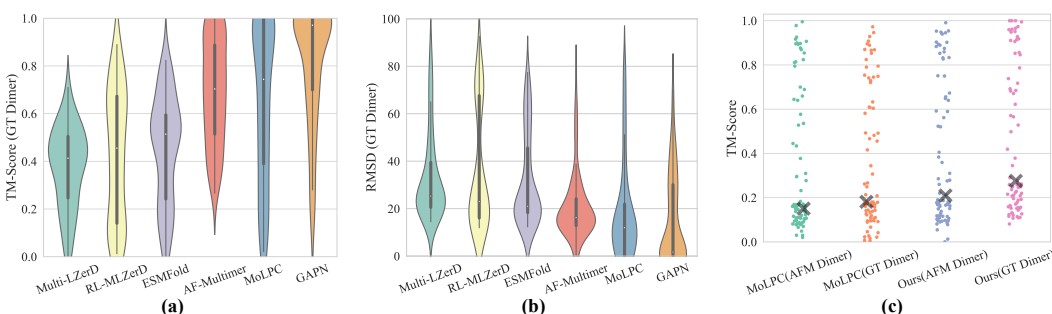

Figure 3: **PCM performance analysis.** **(a).** TM-Score and **(b).** RMSD distributions of all baselines on multimers of $3 \leq N \leq 10$. **(c).** TM-Score distribution of MoLPC and GAPN on large-scale multimers of $11 \leq N \leq 30$, with the median TM-Score marked by ✕.

using 2 GeForce RTX 4090 GPUs. For each sample within the available range, if the testing time exceeds 2 hours or the memory limit is exceeded, its result will be replaced by the optimal value of all baselines for that sample.

**Metrics.** We evaluate the PCM prediction methods by the following metrics: (1) **RMSD**: the root-mean-square deviation of the Cartesian coordinates, measures the distance of the ground-truth and predicted multimer structure at the residue level; (2) **TM-Score**: the proportion of residues that are similar between the ground-truth and predicted multimer structures with values ranging from 0 to 1.

## 4.2 PROTEIN COMPLEX MODELLING

**Performance.** Model performance are summarized in Table 2 and Fig. 3. Our GAPN method achieves state-of-the-art in all ranges of chain numbers. Compared to the second-best baseline with small-scale multimers ($3 \leq N \leq 10$), we achieve improvements of 8.7%, 9.6%, 11.0%, and 12.7% on metrics of mean TM-Score, median TM-Score, mean RMSD and median RMSD, respectively. Moreover, our methods consistently outperform the most advanced assembly-based MoLPC method, without resource-intensive exhaustive search or expensive plDDT information. Fig. 3c shows the comparison of MoLPC and our method for handling large-scale multimers. We can find that although relatively imprecise, AFM dimers can still achieve accurate assembly in practice. This further reinforces the practical significance of the assembly-based setting that we follow. Importantly, our method outperforms MoLPC on large-scale multimers in both settings of AFM and GT dimers. For instance, under the setting of GT dimer, we achieve an improvement of about 31% and 53% in terms of the mean and median TM-Score metrics for multimers of $11 \leq N \leq 30$.

**Visualization.** In Fig. 6, we show two examples with a small ($N = 5$) and a large ($N = 17$) chain number, respectively. These two complexes are both unknown samples, meaning that none of their chains have more than 10% similarity with any chain present in the training set.

**Efficiency.** We evaluate the two types of efficiency of our GAPN method, computational and exploration efficiency, from both the training and inference perspectives. For inference, since the running time of all methods increases with the number of chains, we compare the ratio of running time to chain number (T/N) among different methods. As shown in Table 3, the inference efficiency of our GAPN is about 600 times that of MoLPC. More importantly, our method is not limited by chain numbers, and can typically predict up to 60-chain complexes in less than 2 minutes. We illustrate the training efficiency (exploration efficiency) with the training curves shown in Fig. 4b. We observe that our

Table 3: Inference time.

| Methods | T/N(sec) |
| --- | --- |
| Multi-LZerD | 1235.20 |
| RL-MLZerD | 1644.78 |
| AFM | 357.30 |
| ESMFold | 7.73 |
| MoLPC | 540 |
| GAPN | 0.85 |

GAPN model can converge rapidly and effectively after only $\sim 400$ episodes. This indicates that the model can efficiently explore a vast combination optimization space ($N^{N-2}, 11 \leq N \leq 30$), which also demonstrates the high training efficiency of GAPN (requiring $\sim 1$ hour for a complete training on the whole 6,054 samples).

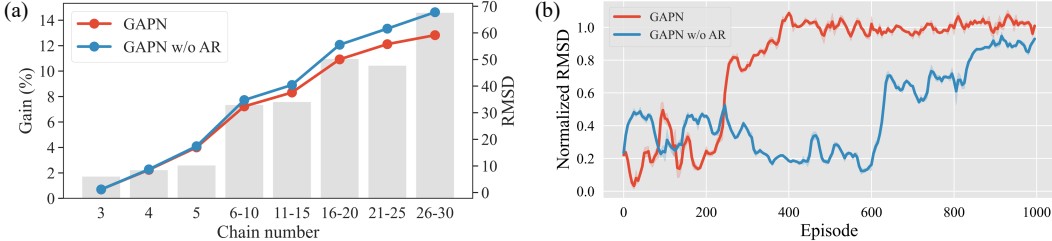

Figure 4: **(a).** The RMSD performance of GAPN and the one without Adversarial reward (AR). The bar chart represents the relative difference between the two for multimers of each specific scale. **(b).** Training curve comparison between our GAPN model and the one without AR. Both models are trained on multimers of $3 \leq N \leq 30$ with GT dimer structures.

## 4.3 ABLATION STUDY

We conduct an ablation study on the whole validation set to explore the importance of our core adversarial reward (AR). Specifically, we call the process of GAPN assembling a multimer an episode and testing the policy network on the validation set after training the policy in each episode. In Fig. 4a, we find that GAPN's performance significantly decreases after removing AR. Importantly, the gain brought by AR will become more significant with the increase of chain number, which indicates that AR can effectively address the problem of knowledge generalization difficulty caused by data scarcity in large-scale multimers. Moreover, we also observe in Fig. 4b that AR is crucial for model training, as it enables rapid convergence of our model.

## 5 CONCLUSION

In this paper, we propose a novel generative adversarial policy network (GAPN) to tackle the problem of Protein Complex Modelling (PCM). We follow the assembly-based setting to predict the docking path of chains and finally obtain the complex structures. GAPN has been demonstrated to efficiently and actively explore the vast combinatorial optimization space with a reinforcement learning (RL) agent. Moreover, we design the domain-specific reward and the adversarial reward (AR) so that the RL agent can acquire knowledge from the current and global perspectives. AR can effectively enhance the receptive field of our model, which is beneficial for the model to generalize knowledge across complexes of different scales. Extensive experiments on the complex dataset show that GAPN achieves significant accuracy and efficiency improvements.

ACKNOWLEDGEMENTS

This work was supported by NSFC Grant No. 62206067, HKUST(GZ)-Chuanglin Graph Data Joint Lab, HKUST-HKUST(GZ) 20 for 20 Cross-campus Collaborative Research Scheme C019 and Guangzhou-HKUST(GZ) Joint Funding Scheme 2023A03J0673.

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

# A  NOTATIONS

We summarize the commonly used notations of our paper in Table 4.

Table 4: A list of commonly used notations.

| Notation | Description |
|---|---|
| $\mathcal{M}$ | The complex. |
| $s_t$ | The state of RL agent at time interval $t$. |
| $c_t$ | Docked chain feature embeddings. |
| $u_t$ | Undocked chain embeddings and $G_r$ denotes all chain embeddings. |
| $G_r$ | All chain embeddings in the complex. |
| $e_i$ | The chain embedding of $M_i$. |
| $a_t$ | The action of RL agent at time interval $t$. |
| $r$ | The reward function of RL agent. |
| $r_t$ | The domain-specific rewards at time interval $t$. |
| $r_{ad}$ | The adversarial rewards. |
| $f_c, f_a$ | Multilayer perceptron. |
| $\pi_\theta$ | The GAPN. |
| $G^r$ | Ground-truth assembly graphs. |
| $r_{ad}$ | The adversarial rewards. |
| $f_c, f_a$ | Multilayer perceptron. |
| $D_\phi$ | Ground-truth assembly graphs. |
| $G_i^f$ | Generated assembly graphs. |
| $p_{data}$ | Ground-truth assembly action sets of complex data with different chains. |
| $x$ | Assembled graph. |
| $V_t, V(\cdot)$ | Value estimated by value network. |
| $H^l$ | The node embedding of the $l - th$ layer. |

# B  DATASET

Our goal is to screen high-quality and non-redundant 3D structural data of complexes. We first downloaded the first biological assembly version of all available complexes released in the PDB database before 2022-6-10 (https://www.rcsb.org/docs/programmatic-access/file-download-services). We have mined a total of 283,346 complexes before the formal data processing. we process and screen multimers according to the following steps:

- We only keep PDB files obtained with X-ray or EM.

- We remove complexes whose NMR structure resolution below 3.0.

- We remove chains with less than 50 residues.

- We remove complexes with an average buried surface area below 500 or NMR structure resolution below 3.0.

- We cluster all individual chains of remaining complexes on 50% similarity with CD-HIT.

- We remove the complexes if the respective clusters of its chains are all more than one member.

- We randomly select samples according to the required number to form the test set.

Please note that different data processing orders may lead to different results. The above process avoids data leakage between the test set and the training set, as well as the occurrence of data redundancy within the training set. We only used a small-sized test set, which was due to the time-consuming nature of running the involved baselines. However, our dataset allows for random partitioning into any test set size, and we can always ensure that there is no data leakage.

| Chain number | Train | Valid | Test |
|---|---|---|---|
| 3 | 1526 | 254 | 20 |
| 4 | 1041 | 174 | 20 |
| 5 | 869 | 183 | 20 |
| 6-10 | 1654 | 276 | 40 |
| 11-15 | 267 | 45 | 20 |
| 16-20 | 206 | 34 | 20 |
| 21-25 | 166 | 28 | 20 |
| 26-30 | 98 | 16 | 20 |
| **Overall** | **6054** | **1009** | **180** |

Table 5: The dataset statistic after complete data processing and filtering.

## C ADDITIONAL EXPERIMENTS

### C.1 KNOWLEDGE GAP BETWEEN CHAIN NUMBERS.

We validated the existence of knowledge gaps between complexes of different sizes (chain numbers) using a reinforcement learning paradigm on our own GAPN model. As shown in the Fig. 5, the horizontal axis represents the number of sizes of complexes we used to train individual models. We tested the well-trained 8 models on a selected dataset containing only 7-chain complexes. The results show that the model trained with the 7-chain complexes is the most powerful to test the 7-chain complexes. Both the TM-Score and RMSD curves show a clear V-shape with the peak or valley at $N = 7$.

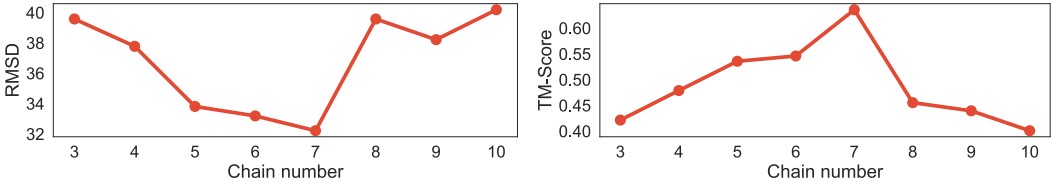

Figure 5: Knowledge gap analysis.

### C.2 PERFORMANCE OF GAPN ON A LARGER-SIZED TEST SET

We partitioned a larger test set according to the release time of the complexes to test our model. However, we are unable to show the results of the baselines due to efficiency limitations. We split the training and test sets using the date of 2005-1-1 as the threshold, with 5530/1710 samples in training/test sets.

The results in Table 6 show that the performance exhibited in large sizes is not significantly different from the performance in Table 2. This indicates that the selected 180 test samples in the main text are representative, and our model does not exhibit a significant decrease in performance due to the smaller training set. Although using AFM as the dimer structure resulted in a noticeable decrease in each metric, the results are still competitive. For example, for complexes with a chain number of about 10, the average TM-Score can reach 0.55.

Table 6: GAPN performance on a large-size test set.

| Methods | Chain numbers $N$ | | | | | | | |
|---|---|---|---|---|---|---|---|---|
| | 3 | 4 | 5 | 6-10 | 11-15 | 16-20 | 21-25 | 26-30 |
| | TM-Score (mean) / RMSD (mean) | | | | | | | |
| GAPN (AFM Dimer) | 0.91 / 1.95 | 0.78 / 6.70 | 0.69 / 16.63 | 0.62 / 38.77 | 0.52 / 46.56 | 0.40 / 56.17 | 0.32 / 62.62 | 0.30 / 66.58 |
| GAPN (GT Dimer) | 0.99 / 0.76 | 0.85 / 4.69 | 0.79 / 15.73 | 0.68 / 33.37 | 0.63 / 38.56 | 0.46 / 51.17 | 0.40 / 52.48 | 0.36 / 58.99 |

## C.3 VISUALIZATION

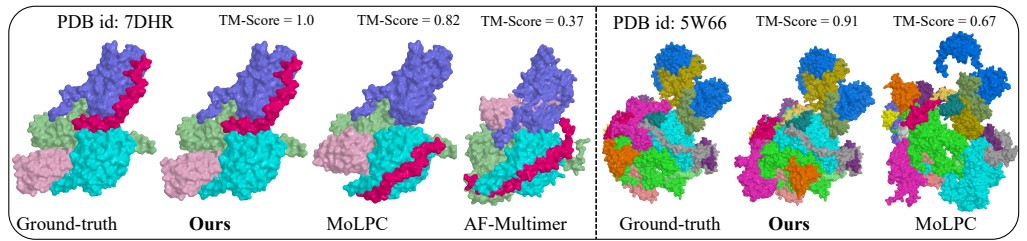

Figure 6: **Visualization of complexes successfully predicted by GAPN.** We show the visual and qualitative results of a small-scale (7DHR, $N = 5$) and a large-scale (5W66, $N = 17$) multimer.

## D HYPERPARAMETERS

We summarize the hyperparameter choices for GAPN in Table 7.

| Hyperparameters | Values |
|---|---|
| Embedding function dimension (input) | 13 |
| GCN layer number | 2 |
| Dimension of MLP in policy network | 13, 32,32 |
| Dimension of MLP in value network | 13,32,64,1 |
| Dimension of GCN in discriminator | 13, 10,10,1 |
| Dropout rate | 0.2 |
| Clip ratio $\epsilon$ | 0.2 |
| Gamma of RL training | 1 |
| Entropy penalty coefficient | 0.01 |
| Max grad norm | 0.5 |
| Batch size | 100 |
| Initial learning rates of policy network, value network, discriminator | $3 \times 10^{-4}, 6 \times 10^{-4}, 3 \times 10^{-4}$ |
| Optimizer | Adam |

Table 7: Hyperparameter choices of GAPN.

## E OVERVIEW OF ACTOR-CRITIC (AC) AND PROXIMAL POLICY OPTIMIZATION (PPO) ALGORITHM

The Actor-Critic (AC) strategy, as proposed by Konda & Tsitsiklis (2000), adeptly integrates strengths from both value-driven Mnih et al. (2013) and policy-driven Sutton et al. (1999) approaches. The framework posits two essential components: a critic $V_{\pi_\theta}$ and an actor $\pi_\theta$.

The critic, $V_{\pi_\theta}$, essentially embodies the value-oriented approach. It assesses and predicts the potential worth of a present state during the learning phase. A core objective is to curtail the Temporal Difference (TD) $\delta_t$ error to ensure a refined valuation of the current state:

$$\delta_t = (r(s_t, a_t) + V_{\pi_\theta}(s_{t+1}) - V_{\pi_\theta}(s_t))^2. \tag{8}$$

Conversely, the actor, $\pi_\theta$, symbolizes the policy-centric method. By engaging actively with the given environment, it generates actions in alignment with the prevailing policy. With the help of the advantage function $A_{\pi_\theta}$, the actor, $\pi_\theta$, enjoys enhanced stability over conventional policy gradient techniques Williams (1992):

$$A_{\pi_\theta}(s_t, a_t) = r(s_t, a_t) + V_{\pi_\theta}(s_{t+1}) - V_{\pi_\theta}(s_t). \tag{9}$$

Subsequent to this, updates to the actor, $\pi_\theta$, are made via the function $J(\theta)$:

$$\nabla J(\theta) = E(\nabla_\theta log\pi_\theta(s_t, a_t) A_{\pi_\theta}(s_t, a_t)), \tag{10}$$

where $E$ signifies the expected value. The PPO algorithm adds policy update constraints on the basis of AC to make the training process more stable.

