# OpenReview forum: "Deep Reinforcement Learning for Modelling Protein Complexes"
_ICLR.cc/2024/Conference — ICLR 2024 poster_

### Official Review · Reviewer_rNSL · 2023-10-29

**Soundness:** 3 good
**Presentation:** 3 good
**Contribution:** 2 fair
**Rating:** 6
**Confidence:** 5

**Summary:**

The paper introduces GAPN, a model designed to predict docking complexes through the application of reinforcement learning. The agent is tasked with determining the optimal docking path, subsequently utilizing precomputed dimer structures to assemble the entire complex. The paper provides an extensive comparative analysis with established methods such as Multi-LZerD, RL-MLZerD, AF-Multimer, ESMFold, and MoLPC, demonstrating GAPN's superior performance across various protein chain number ranges, as evidenced by metrics like TM-Score and RMSD.

**Strengths:**

1. The paper is articulate, well-structured, and easy to follow.

2. The authors have provided a detailed description of the RL setup, including the state, action, transition, and reward components, which are crucial for understanding the methodology.

**Weaknesses:**

1. The model primarily focuses on predicting the docking path, relying on precomputed dimer structures for the completion of the docking task. This reliance on precomputed structures potentially limits the model’s applicability in real-world scenarios, as it is sensitive to the quality of these precomputed dimers.

2. The protein docking process is simplified to a sequential decision-making problem in the proposed method. Each step in this process is determined by a precomputed dimer. While the precomputed dimers (utilizing AFM or ESMFold) effectively capture bi-interactions, they may not adequately represent the complex interactions that can occur across different chains in a protein complex in real-world settings.

3. There appear to be inconsistencies in some of the comparative results presented. GAPN has the advantage of accessing precomputed/prefetched dimer structures, reducing its task to merely finding a docking path. This setup makes the inference time incomparable to other baselines like AFM and ESMFold, which adopt an end-to-end approach for predicting complex structures. Consequently, the inference speedup showcased in the tables may not translate to tangible benefits in practical applications.

**Questions:**

1. Could you please clarify the source of the dimer structures used by GAPN in Table 2? If GT-Dimer is utilized, it implies that GAPN has direct access to the ground truth relative positions of different chains, which could potentially skew the results and present a misleading improvement.

2. In Section 4.1, the paper mentions that all possible pairs of chains are precomputed or fetched. Could you provide details on the average time spent in this step and the average number of dimer structures generated?

3. In Table 1, it is indicated that GAPN does not optimize the dimer structure. Could this lead to error accumulation, especially if some precomputed dimer structures significantly deviate from the ground truth dimer structures?

---

> ### Author Response · Authors · 2023-11-20
> **Response to the Reviewer rNSL (1/3)**
>
> We sincerely thank the reviewer **rNSL** for appreciations and valuable comments on our paper. To address reviewer rNSL's questions, we provide pointwise responses below.
>
> **[Cons1. Reliance on precomputed dimer structures]** Thanks for this valuable comment. **We sincerely acknowledge that the quality of precomputed dimers is important for our method, and this situation is also present in MoLPC [R1] (our baseline).** In fact, our intuition is that given the data of dimers (2 proteins) is rich while the data of protein complex (>> 2 proteins) is sparse, we would like to transfer the knowledge of precomputed dimers to predict high quality protein complex.
>
> Next, we summarize why the applicability of our method in the real-world scenarios is satisfactory.
> 1. We have considered using AFM and ESMFold to precompute dimer structures, and with either of them, our framework can significantly outperform other baselines.
> 2. When using AFM and ESMFold to precompute dimer structures, our method does not require monomer (chain) structures as input.
> 3. During inference, our method also does not require monomer (chain) structures to predict the docking path.
> 4. In the future, there may be some dimer structure prediction methods that can adapt our framework for further improvement.
>
>
> **To address your concerns, we provide the detailed results with AFM-produced dimers and explore an alternative method, ESMFold, to precompute dimer structures.**
>
>
> All results will be added to Table 2 (the main experiment table) in our revised manuscript.
> * MoLPC(GT): the performance of MoLPC with GT dimers.
> * MoLPC(AFM): the performance of MoLPC with AFM-produced dimers.
> * MoLPC(ESMFold): the performance of MoLPC with ESMFold-produced dimers.
> * GAPN(GT): the performance of ours with GT dimers.
> * GAPN(AFM): the performance of ours with AFM-produced dimers.
> * GAPN(ESMFold): the performance of ours with ESMFold-produced dimers.
>
> Metric: mean TM-Score.
> The best and second best, other than GAPN(GT) and MoLPC(GT), is **bolded** and `highlighted`, respectively.
>
> | Chain Number   |  3   |  4   |  5   | 6-10 | 11-15 | 16-20 | 21-25 | 26-30 |
> | --------- |:--:|:----:|:----:|:----:|:-----:|:-----:|:-----:|:-----:|
> | Multi-LZerD    | 0.54 | 0.49 | 0.40 | 0.24 |  --   |  --   |  --   |  --   |
> | RL-MLZerD      | 0.62 | 0.47 | 0.49 | 0.31 |  --   |  --   |  --   |  --   |
> | AF-Multimer    | 0.79 | 0.70 | `0.67` | **0.64** | 0.39  |  --   |  --   |  --   |
> | ESMFold        | 0.73 | 0.52 | 0.44 | 0.36 |  --   |  --   |  --   |  --   |
> | MoLPC(AFM)     | 0.76 | 0.63 | 0.56 | 0.50 | 0.36  | 0.36  | `0.30`  | 0.29  |
> | **GAPN(AFM)**      | `0.81` | **0.75** | **0.70** | `0.61` | **0.52**  | **0.44**  | **0.38**  | `0.33`  |
> | MoLPC(ESMFold) | 0.79 | 0.60 | 0.52 | 0.47 | 0.38  | `0.38`  | 0.29  | 0.26  |
> | **GAPN(ESMFold)**  | **0.83** | `0.73` | 0.64 | 0.59 | `0.42`  | **0.44**  | **0.38**  | **0.38**
> | MoLPC(GT)      | 0.89 | 0.71 | 0.66 | 0.60 | 0.45  | 0.36  | 0.34  | 0.29 |
> | GAPN(GT) | 0.95 | 0.81 | 0.79 | 0.65 | 0.62 | 0.48 | 0.38 | 0.36 |
>
> **We have the following observations:**
> * Using dimers produced from AFM and ESMFold, our model can consistently outperform AF-Multimer (a baseline without precomputed dimer).
> * Only MoLPC and our method can handle cases with a large number of chains ($N>15$).
> * For our method, the performance with AFM-produced dimers and ESMFold-produced dimers is comparable.
>
> More importantly, two existing dimer prediction methods that do not require given monomer (chain) structures have been verified to adapt well to our model. Among them, ESMFold has very high efficiency (about 0.6 seconds to predict a dimer structure).
>
> As a result, we believe this demonstrates the applicability of our method in real-world scenarios. In the future, perhaps more dimer structure prediction methods will be verified to perform excellently within the GAPN framework.

---

> > ### Author Response · Authors · 2023-11-20
> > **Response to the Reviewer rNSL (2/3)**
> >
> > **[Cons2. Bi-interaction of AFM]** Thanks for the valuable insights. The problem (complex structure prediction) itself involves optimizing two modules, which are the docking path and the dimer structures. Ideally, both the decision for docking path and the prediction of dimer structures should be "flexible", taking into account the environment of the entire complex. In this paper, we do not optimize the dimer structures explicitly, for the following two reasons.
> > 1. We sincerely believe that our model can be "flexible", meaning it can capture the entire complex's environment. A major characteristic of our reinforcement learning (RL) framework is that it can model the problem as a Markov Decision Process, **allowing the RL agent to consider the impact of current decision on future gains (long-term benefits),** rather than simply considering the currently state (assembled complex). Therefore, during training, although we do not directly optimize dimer structures, our model adjusts the docking path based on the given dimer structures and environment of the entire complex.
> > 2. Considering the optimization of dimer structures is a non-trivial problem beyond our RL framework. Besides, it is evident that introducing this optimization space (flexibility) may reduce computational efficiency with little performance improvement **[R2]**.
> >
> > Fortunately, within our setting, we can achieve significantly better performance with predicted docking paths and AFM dimers compared to the AF-Multimer baseline. This observation also aligns with that mentioned in MoLPC **[R1]**, which concludes that accurate docking path, along with precomputed dimer structures, can together handle this task well. **We believe that dual awareness of the multi-chain environment may further improve performance by optimizing both the dimer structure and docking paths.** We sincerely appreciate that the reviewer's suggestion may imply a promising future research direction.
> >
> > **[Cons3. Total Inference time]** Thanks for your insightful comments. Since our method is essentially a framework that can be applied to any dimer type, in the original manuscript, we only showed the running time of the docking path prediction step. **Based on the reviewer's suggestion, we sincerely believe that to ensure a fair comparison, it is necessary to consider the total running time in real-world scenarios. Therefore, we take this factor into account and provide more details.**
> >
> > Total running time for each testing complexes with $3 \leq N \leq 10$ :
> > | Time(min) | Docking path | Dimer pre-computation | Total for each |
> > |---|:---:|:---:|:---:|
> > | Multi-LZerD | -- | -- | 160.76 |
> > | RL-MLZerD | -- | -- | 179.93 |
> > | AF-Multimer | -- | -- | 147.52 |
> > | ESMFold | -- | -- | 1.78 |
> > | MoLPC(AFM) | 9.71 | 225.60 | 235.31 |
> > | **GAPN(AFM)** | 0.09 | 82.12 | 82.21 |
> > | **GAPN(ESMFold)** | 0.09 | 0.13 | 0.22 |
> > | **GAPN(GT)** | 0.09 | -- | 0.09 |
> >
> > The running time of the AF-Multimer baseline and baselines involving AFM-produced dimers is relatively long. However, when actually implementing it, multiple GPUs can be used in parallel to significantly reduce the running time of AFM for processing multiple dimer structures.
> >
> > **Even when using AFM-produced dimers, our method is still significantly faster than the AF-Multimer baseline. To explain this, we provide some intuitive reasons below.**
> >
> > We assume the sequence length of each chain is $L$. For an $N$-chain protein complex, the input length for AF-Multimer (a baseline without precomputed dimer) would be $NL$. For GAPN(AFM), we use AFM to predict $N-1$ dimer structures, and for each of them, we input a sequence of length $2*L$ to the AFM. As the computational complexity of Transformer's self attention component is $O(L^2)$ **[R3,R4]**, thus the running time ratio of AF-Multimer to GAPN(AFM)'s is about $\frac{N^2L^2}{(N-1){(2L)}^2}=\frac{N^2}{4(N-1)}$. Therefore, as long as $N \geq 3$, this ratio will be greater than $1$. The larger the $N$, the greater the ratio.
> >
> > **We will add these results and details to the Appendix F in our revised manuscript.**

---

> > > ### Author Response · Authors · 2023-11-20
> > > **Response to the Reviewer rNSL (3/3)**
> > >
> > > **[Q1 Dimer structure prediction methods]** Thanks for the valuable question. In the original Table 2, both MoLPC and GAPN were assembled using ground-truth dimer structures.  We will use the results in **[cons1]** to update Table 2 in our revised manuscript. Overall, it can be seen that GAPN based on AFM-produced dimers or ESMFold-produced dimers still outperforms other baseline methods.
> > >
> > > **[Q2 Overall running time]** Thanks for the valuable question. Please kindly refer to our response to **[Cons3]**, where we report the running time for both docking path prediction and dimer structure computation. Besides, we will revise the results in Table 1 by considering the entire running time of each method.
> > >
> > > We respectfully clarify that dimer prediction is only required during the inference stage. **For an $N$-chain protein complex, we need to precompute $N-1$ dimer structures.** As an example, we consider a protein complex with $4$ chains "A,B,C,D". Our model may output the path **C**$\rightarrow$C-**D** $\rightarrow$ C-**B** $\rightarrow$ D-**A** (the newly added chain is **bolded**). Therefore, we only need to obtain the structures of the $3$ dimers: CD, CB, and DA.
> > >
> > > **[Q3 Error accumulation]** Thanks for your insightful question. **We strongly agree that addressing error accumulation during assembly is important in this paper.** Instead of optimizing dimer structures, we adopt a Markov Decision Process (MDP) modeling approach as an alternative solution to alleviate this issue. MDP is well-known for its focus on **long-term benefits**, that is, obtaining maximum rewards in a series of future decisions. In this case, even if some decisions in the process inevitably lead to errors, MDP tries to minimize the final (future) error as much as possible.
> > >
> > > **We sincerely appreciate all comments of the reviewer, which greatly helps improve the quality of our paper.**
> > >
> > > **[R1]** Bryant P, Pozzati G, Zhu W, et al. Predicting the structure of large protein complexes using AlphaFold and Monte Carlo tree search. Nature communications, 2022, 13(1): 6028.
> > >
> > > **[R2]** Aderinwale T, Christoffer C, Kihara D. RL-MLZerD: Multimeric protein docking using reinforcement learning. Frontiers in Molecular Biosciences, 2022, 9: 969394.
> > >
> > > **[R3]** Vaswani A, Shazeer N, Parmar N, et al. Attention is all you need. Advances in neural information processing systems, 2017, 30.
> > >
> > > **[R4]** Beltagy I, Peters M E, Cohan A. Longformer: The long-document transformer. arXiv preprint arXiv:2004.05150, 2020.

---

> > > > ### Comment · Reviewer_rNSL · 2023-11-22
> > > >
> > > > Thank you for your detailed response. While I still have reservations about the foundational assumptions of your approach (specifically, the modeling of the docking problem using precomputed dimer structures and optimizing a docking path), I acknowledge the thoroughness of your rebuttal. In recognition of your effort and the depth of your explanations, I have decided to raise my rating to 6.

---

> ### Author Response · Authors · 2023-11-23
> **Thanks for the reviewer**
>
> Thank you for your positive feedback. We appreciate your valuable suggestions and the discussions.

---

### Official Review · Reviewer_554U · 2023-10-31

**Soundness:** 3 good
**Presentation:** 2 fair
**Contribution:** 3 good
**Rating:** 6
**Confidence:** 3

**Summary:**

The authors proposed a novel deep reinforcement learning model, named GAPN, to efficiently explore the vast combinatorial optimization space in the protein complex modelling (PCM) prediction problem to find the optimal assembling action. GAPN allows active exploration of a vast space of potential solutions to strike a balance between trying new actions and sticking with known good actions, ensuring it avoids local optima. Additionally, an adversarial reward function incorporating global assembly rules enhances the model's learning process and generalization abilities.

**Strengths:**

1. The paper is well-organized and easy to follow.
2. Experimental results demonstrates the effectiveness of proposed GAPN in both prediction accuracy and efficacy.

**Weaknesses:**

This idea is not new, and the models they used are all well established.

**Questions:**

N.A.

---

> ### Author Response · Authors · 2023-11-20
> **Response to the Reviewer 554U (1/1)**
>
> We sincerely appreciate the reviewer's valuable time and suggestions. We address the reviewer's concern with the following response.
>
> **[Cons. Idea is not new]** In this paper, we propose a reinforcement learning (RL) framework for the complex structure prediction problem. We sincerely clarify that the novelty of this paper is twofold.
> 1. We model the complex structure prediction problem as a Markov Decision Process for long-term benefits. This differs from the advanced baseline methods **[R1,R2]**, which focus on greedily ensuring locally optimal assembly. This modeling ensures high exploration efficiency w.r.t. the vast action space and achieves excellent experimental results.
> 2. The design of a global view reward specifically for protein complex problems effectively addresses critical challenges of the task, which have been rarely mentioned in existing research. Overall, the introduction of the generative adversarial reward addresses the challenges of both the gap in knowledge between protein complexes of varied scales (chain numbers) and the scarcity of data for large-scale protein complex.
>
> **[R1]** Bryant P, Pozzati G, Zhu W, et al. Predicting the structure of large protein complexes using AlphaFold and Monte Carlo tree search. Nature communications, 2022, 13(1): 6028.
>
> **[R2]** Esquivel‐Rodríguez J, Yang Y D, Kihara D. Multi‐LZerD: multiple protein docking for asymmetric complexes. Proteins: Structure, Function, and Bioinformatics, 2012, 80(7): 1818-1833.

---

> > ### Comment · Reviewer_554U · 2023-11-22
> >
> > Thank you for the response and my concern is addressed to some extend. Thus, I increased the confidence score from 2 to 3.

---

> ### Author Response · Authors · 2023-11-23
> **Thanks for the reviewer**
>
> Thank you for your positive feedback. We appreciate your valuable suggestions and the discussions.

---

### Official Review · Reviewer_Sjm2 · 2023-11-01

**Soundness:** 2 fair
**Presentation:** 2 fair
**Contribution:** 2 fair
**Rating:** 6
**Confidence:** 3

**Summary:**

Paper is interested in protein folding. Given a protein, i.e., a "graph" of amino acids where each amino acid is a sequence of "residues", paper would like to output the 3D structure of the protein (i.e., the cartesian xyz coordinates of each residue), subject to rotation and translation invariance. The paper achieves that by repeatedly attaching amino acids to one another. Specifically, if a protein consists of $N$ amino acids, then their algorithm runs for $N - 1$ iterations. At each iteration, a pair of amino acids will be paired (the choice of the pair is output by their RL policy, with softmax activation). Their algorithm allows on running on larger proteins (e.g., >20 amino acids)

-------------

# Update

I read the author's response. Accordingly, I am raising my score

**Strengths:**

# Problem domain
* The paper is an important problem domain: protein folding. Accurate protein folding could imply faster discovery of drugs, especially given the "virus era" erupted by COVID-19.
* The problem comes with some interesting challenges, specifically, models should be in(/equi)variant to rotation and translation.

# Enabling folding of larger proteins

In my understanding (per paper text), the earlier methods either take too long to simulate folding for larger proteins (e.g., >9 chains) or give bad accuracy. The intent of the paper is to fill that gap.

# Dataset

They contribute a dataset containing non-redundant complexes. This is useful to test the generalization of protein modeling.

**Weaknesses:**

# Missing primer / prelim
It would be nice if you give some 4-line summary of terminology (even if brief). The ICLR audience might not be familiar with concepts like "docking" and "dimer".

# Adversarial reward Eq.3

The motivation and implementation of Eq.3 are ambiguous. It says that $p_{data}(x)$ is "the underlying distribution of ground-truth assembly action set". Can you give details to the support of the distribution? Is it actually "pairs of amino acid indices"? Do you even have that information in the dataset? I thought you only have protein chains (not set of pairs of indices). Does every protein become $N-1$ entries in $p_{data}$. Importantly, does the order matter of how you order these edges? Perhaps some amino acid $C$ wouldn't bind to $B$ but would bind (from left) to $B$-$A$.

# GNN

In Equation 4, the aggregation is done by $D^- U D^-$. Can you please explain function "AGG" or remove it, if it was a typo?

# Clarity
* Second page, around the middle: "Balance between exploration and exploitation". Can you please elaborate in the text?

**Questions:**

In addition to the points brought-up in "Weaknesses", above, I have the following questions:

Q1. where does $\theta_{old}$ in Eq.6 and Eq.7 come from?

---

> ### Author Response · Authors · 2023-11-20
> **Response to the Reviewer Sjm2 (1/2)**
>
> We sincerely thank reviewer **Sjm2**’s valuable time and comments. We provide point-wise responses below.
>
> **[Cons1. Terminology description]** Thank you for your comments and for bringing this issue to our attention. To address it, we will provide a more explicit explanation of terminology such as docking, dimer and complex. Our revised manuscript PDF will include all these updates.
>
> Here, we respectfully provide the descriptions to the reviewer. In the main body of our paper, we mostly use the terms **protein** and **chain** exchangeably. For **"Docking"**, it refers to the action that involves any two proteins, where one protein is transformed to a new position to interact (physically contact) with another protein in space. In other words, "docking" results in the formation of a protein complex where the two protein chains finally come into contact with each other. **"Dimer"** refers to a protein complex with **two chains**, which is the result of docking of two protein chains. **"Protein complex"** refers to a protein group containing **two or more chains**. A protein complex with $N$ chains is formed as the result of $N-1$ docking actions.
>
> **[Cons2. Explanation of Eq.3]** We really appreciate the reviewer's detailed comments. Firstly, we apologize for any confusion caused by the term "ground-truth assembly graphs" in our paper. **To clarify, in the revised manuscript, we will modify the sentence to:** "*We search for the assembly graphs that specify true protein complexes. $p_{data}$ refers to the underlying distribution of data for such a set of assembly graphs.*"
>
> First of all, we would like to clarify that there is no amino acid involved in our formulation. Given a set of $N$ protein chains (a protein chain consists of a set of amino acid), we want to predict the best possible protein complex with $N$ chains. However, due to the large computational space, we use "assembly graphs" to merge possible equivalent protein complex. In other words, an "assembly graph" specifies a protein complex in which we pay no attention to the detailed "docking" order. Based on different "assembly graphs", we can specify different protein complexes in which some protein complexes are true and some are not.
>
> * **Details about $p_{data}$: To further illustrate, we take an example of a 3-chain (indices: A, B, and C) complex in the training set.** We pre-compute the embedding (vector) with each chain's amino acid sequence. These embeddings then serve as the node attributes of this 3-node assembly graph. The edges of the selected assembly graph are A-B and B-C. With such an assembly graph, we can **truly** assemble the corresponding protein complexes. Upon this assembly graph formulation, we expect these ones w.r.t true protein complexes to have large likelihood while others to have small likelihood. And this distribution is exactly $p_{data}$.
> * **About pairs of indices: We respectfully clarify that $p_{data}$ is represented by the protein (node) embeddings and assembly action (edges) rather than "pairs of indices".** In Figure 1, we use indices "A,B,C,D" to distinguish each protein. As shown in Eq.4, each assembly graph drawn from $p_{data}$ is input into the discriminator (i.e., Graph Neural Networks) for representation computation. The information of an assembly graph involved in the computation includes its node embeddings and all edges, but not the node indices. Therefore, the indices can be specified arbitrarily as long as it is distinguishable, such as "D,E,F" or "1,2,3".
> * **Every graph becomes $N-1$ entries:** For an $N$-chain protein complex, its assembly graph specifying the true protein complex contains $N$ nodes and $N-1$ edges. Every node has an embedding vector.
> * **Edge order:** In our setup, as long as the set of assembly actions (edges) is determined, the final result is unique. Our discriminator $D_\phi$ will evaluate whether the set of edges is reasonable as a whole.  **Therefore, edge order dose not effect the output of discriminator for the current assembly graph.**
>
> **[Cons3. The "AGG" notation]** Thank you very much for your suggestion. **To clarify, we will revise it using a more commonly used notation in the field of graph learning, namely ReadOut.** This notation refers to the process of **pooling** node embeddings into a graph-level representation. By using the ReadOut function, graphs with different numbers of nodes can be distinguished by the discriminator.

---

> > ### Author Response · Authors · 2023-11-20
> > **Response to the Reviewer Sjm2 (2/2)**
> >
> > **[Cons4. Exploration and exploitation]** Thanks for the comments regarding this issue. **We will include relevant explanations in Appendix E of the revised manuscript.**
> >
> > **Exploration** involves trying new actions to discover their effects, which is essential for acquiring new knowledge about the environment. Without exploration, an agent might never find the most rewarding actions. **Exploitation**, on the other hand, involves using the current knowledge to choose actions that yield the highest reward. An agent that never exploits this knowledge will fail to maximize its rewards, as it will always be trying new things instead of using what it has learned to its advantage.
> >
> > **In the context of reinforcement learning, both of exploration and exploitation need to be satisfied to handle large action spaces ($N^{N-2}$ in this paper).** Effective RL algorithms for large action spaces are designed to find a balance between these two approaches, often adjusting their strategy as they accumulate more knowledge about the environment.
> >
> > **[Q1 Abotu $\theta_{\text{old}}$]** Thanks for the detailed suggestion. In this paper, $\theta_{\text{old}}$ represents the parameters of the policy at a previous iteration or time step. These parameters are part of a policy function, such as a neural network, which determines the agent's actions based on its observations of the environment. The role of $\theta_{\text{old}}$ is crucial when updating the policy to improve performance. As we mentioned in the below of Eq.7 in the manuscript, for algorithms like Proximal Policy Optimization (PPO), $\theta_{\text{old}}$ is used as a reference to ensure that the updated policy does not deviate too much from the previous policy, thus maintaining a certain level of stability in learning.
> > **To provide further clarity, we will add an explanation about $\theta_{\text{old}}$ in the text below Eq. 7.**

---

> ### Author Response · Authors · 2023-11-23
> **We are willing to have a further discussion, especially about  the clarity of our paper**
>
> Dear reviewer,
>
> The discussion period is almost ending. Could you please confirm whether our responses have alleviated your concerns? If you have further comments, we will be happy to discuss them. For your interest, we  clarify the key information on methodology and terminology description. Please refer to the details in the responses for you.
>
> Thank you very much!

---

> > ### Author Response · Authors · 2023-11-23
> > **A friendly reminder for discussion**
> >
> > Dear Reviewer **Sjm2**,
> >
> > We hope this message finds you well. The rebuttal phase ends today and we have not yet received feedback from you. We believe that we have addressed all of your previous concerns. We would really appreciate that if you could check our response and revised manuscript.
> >
> > Looking forward to hearing back from you.
> >
> > Best Regards,
> >
> > The Authors of Paper 1707

---

### Official Review · Reviewer_MBaj · 2023-11-06

**Soundness:** 3 good
**Presentation:** 3 good
**Contribution:** 3 good
**Rating:** 6
**Confidence:** 3

**Summary:**

This paper proposes a generative adversarial policy network (GAPN) for modeling large protein complexes through sequential assembly of individual protein chains. GAPN uses reinforcement learning to efficiently explore the vast combinatorial search space and find optimal assembly actions. The adversarial reward function in GAPN incorporates global assembly knowledge from complexes of various sizes, enabling it to generalize despite limited training data. Overall, GAPN achieves state-of-the-art accuracy in predicting structures of protein complexes across a wide range of chain numbers with significant speed-up.

**Strengths:**

This work integrates reinforcement learning, graph neural networks, and adversarial training to tackle the challenging problem of protein complex modeling, which is novel. In detail, the authors identify the key issues of huge search space and lack of generalization across protein complexes of different sizes. The proposed GAPN framework well addresses these challenges through policy-based active search and an adversarial reward function that encodes global assembly knowledge. The graph representation of protein chains and complexes is well-motivated and fits naturally with the assembly actions. The experiments comprehensively evaluate performance over a range of complex sizes, convincingly demonstrating GAPN's accuracy and efficiency advantages when compared to existing baselines. The ablation studies validate the benefits of the adversarial reward.

**Weaknesses:**

It would be more helpful and intuitive to provide the assembly process of GAPN and MoLPC for the examples shown in Figure 4.

For the efficiency analysis, it would be better to also theoretically analyze the exploration complexity and empirically analyze the relationship between efficiency and chain number N.

**Questions:**

The authors mention that they apply two types of dimer structures, the ground-truth dimers (GT Dimer) and dimers predicted by AlphaFold-Multimer (AFM Dimer). In Table 6, the GAPN results using either of them are noted and provided. Are the results in Table 2 achieved by using both or one of them?

---

> ### Author Response · Authors · 2023-11-20
> **Response to the Reviewer MBaj (1/2)**
>
> We appreciate the valuable time and suggestions from the reviewer **MBaj**, and we also thank the reviewer for recognizing the contributions and novelty of our paper. To address these concerns, we provide point-wise responses below.
>
> **[Cons1. Assembly Process]** We sincerely thank the reviewer for this insightful suggestion.
>
> **We strongly agree that providing the assembly process is necessary and important for explaining the effectiveness of our method.** As our model considers modeling the problem as a Markov Decision Process (MDP), the reinforcement learning (RL) agent will focus on the long-term impact of current decisions on future rewards (long-term benefits). This means that the assembly process of our method may not greedily maximize the current TM-Score, which is evident in the example 5W66 we have provided in Figure 4.
>
> We will add visualizations of the assembly process for Figure 4's cases in Appendix G.
>
> Here, we provide the assembly process and the current rewards (TM-Score) for each step.
>
> $\rightarrow$: the assembly process; N-**B**: chain **B** (**newcomer**) is docked to chain N (already docked).
>
> * Complex 7DHR with chain indices A, B, G, N, R.
>
> Asssmbly process for 7DHR: **N**$\rightarrow$ N-**B**$\rightarrow$ N-**A** $\rightarrow$ A-**R** $\rightarrow$ B-**G**
> * Complex 5W66 with chain indices A, B, C, D, E, F, G, H, I, J, K, L, M, N, P, Q.
>
> Asssmbly process for 5W66: **M**$\rightarrow$ M-**F** $\rightarrow$ M-**A** $\rightarrow$ A-**E** $\rightarrow$ E-**H** $\rightarrow$ F-**C** $\rightarrow$ A-**K** $\rightarrow$ M-**N** $\rightarrow$ N-**B** $\rightarrow$ B-**J** $\rightarrow$ B-**Q** $\rightarrow$ Q-**P** $\rightarrow$ B-**G** $\rightarrow$ G-**D** $\rightarrow$ B-**I** $\rightarrow$ J-**L**
>
> Calculated TM-Scores in order for 5W66: 1.0$\rightarrow$ 1.0$\rightarrow$ 1.0$\rightarrow$ 1.0$\rightarrow$ `0.93`$\rightarrow$ `0.76`$\rightarrow$ `0.72`$\rightarrow$ 0.80$\rightarrow$ 0.83$\rightarrow$ 0.83$\rightarrow$ 0.86$\rightarrow$ 0.88$\rightarrow$ 0.90$\rightarrow$ 0.92$\rightarrow$ 0.92$\rightarrow$ `0.91`
>
> It can be observed that our method starts making mistakes from the 5-th step (highlighted as `0.93`), but the **majority of subsequent actions increase the reward.** This indicates that our method does not greedily choose the optimal solution for the current action, but may make early mistakes to ensure the maximization of future rewards. For 5W66, if the correct decision had been forcefully (manually) made at the 5-th step, the final TM-Score that GAPN achieves will been 0.83, which explains the effectiveness of our model.
>
> **[Cons2. Exploration efficiency]** Thanks for the valuable suggestion.
>
> **Empirical analysis:** We present numbers of episodes required for convergence when training on datasets with different $N$ (chain numbers).
> | Chain Number $N$  |  3 |  4 |  5  | 6 | 7 | 8 | 9 | 10 |
> | ---------- |:----:|:----:|:----:|:----:|:-----:|:-----:|:-----:|:-----:|
> | Convergence episode    | 119 | 114 | 319 | 424 |  575   |   680  | 892  | 1254    |
>
> We will include the detailed convergence curves to our revised manuscript later for better presentation. It can be observed that $N$ has a significant impact on the convergence speed of our model, which is due to the increase in its total action space $N^{N-2}$. Importantly, we find that our model often converges within a small number of episodes. For example, when $N=10$, the entire action space is $10^8$, and our model converges within 1254 episodes.
>
> **Theoretical analysis:** We have drawn from some existing studies **[R1,R2]** to demonstrate that the Proximal Policy Optimization (PPO) framework instantiated with neural networks can have local or global convergence guarantee under certain assumptions. We will add these theorems to the revised manuscript.
>
>
> Kindly note that all reinforcement learning (RL) models using **deep neural networks** cannot rely on theory to absolutely guarantee convergence, as RL models in real-world practice like our protein complexes assembly application may not meet the strict assumptions and may be affected by many factors **[R3,R4]**, such as learning rate, training set, network structure design, discount factor, replay buffer size and reward function design. However, the successful application of the actor-critic algorithm like PPO  in a large number of practical problems with vast action spaces **[R5,R6]** indicates that it can usually converge efficiently to good policies in practice.

---

> > ### Author Response · Authors · 2023-11-20
> > **Response to the Reviewer MBaj (2/2)**
> >
> > **[Q1 Types of dimer structures]** Thanks for this valuable question and we apologize for the lack of description in the original Table 2.
> >
> > We first respectfully clarify that the performance of both MoLPC and our method in the original Table 2 is based on ground-truth dimer structures. To further address this issue, we will add results based on AFM-produced dimers to Table 2, and we newly consider another type of dimer, i.e., using ESMFold to predict dimer structures.
> >
> > All results will be added to Table 2 in our revised manuscript.
> > * MoLPC(GT): the performance of MoLPC with GT dimers.
> > * MoLPC(AFM): the performance of MoLPC with AFM-produced dimers.
> > * MoLPC(ESMFold): the performance of MoLPC with ESMFold-produced dimers.
> > * GAPN(GT): the performance of ours with GT dimers.
> > * GAPN(AFM): the performance of ours with AFM-produced dimers.
> > * GAPN(ESMFold): the performance of ours with ESMFold-produced dimers.
> >
> > Metric: mean TM-Score.
> > The best and second best, other than GAPN(GT) and MoLPC(GT), is **bolded** and `highlighted`, respectively.
> >
> > | Chain Number   |  3   |  4   |  5   | 6-10 | 11-15 | 16-20 | 21-25 | 26-30 |
> > | -------------- |:----:|:----:|:----:|:----:|:-----:|:-----:|:-----:|:-----:|
> > | Multi-LZerD    | 0.54 | 0.49 | 0.40 | 0.24 |  --   |  --   |  --   |  --   |
> > | RL-MLZerD      | 0.62 | 0.47 | 0.49 | 0.31 |  --   |  --   |  --   |  --   |
> > | AF-Multimer    | 0.79 | 0.70 | `0.67` | **0.64** | 0.39  |  --   |  --   |  --   |
> > | ESMFold        | 0.73 | 0.52 | 0.44 | 0.36 |  --   |  --   |  --   |  --   |
> > | MoLPC(AFM)     | 0.76 | 0.63 | 0.56 | 0.50 | 0.36  | 0.36  | `0.30`  | 0.29  |
> > | **GAPN(AFM)**      | `0.81` | **0.75** | **0.70** | `0.61` | **0.52**  | **0.44**  | **0.38**  | `0.33`  |
> > | MoLPC(ESMFold) | 0.79 | 0.60 | 0.52 | 0.47 | 0.38  | `0.38`  | 0.29  | 0.26  |
> > | **GAPN(ESMFold)**  | **0.83** | `0.73` | 0.64 | 0.59 | `0.42`  | **0.44**  | **0.38**  | **0.38**
> > | MoLPC(GT)      | 0.89 | 0.71 | 0.66 | 0.60 | 0.45  | 0.36  | 0.34  | 0.29 |
> > | GAPN(GT) | 0.95 | 0.81 | 0.79 | 0.65 | 0.62 | 0.48 | 0.38 | 0.36 |
> >
> > Based on these results, we can see that our model consistently outperforms other baselines when using dimers produced from both AFM or ESMFold. Besides, only MoLPC and our method can handle cases with a large number of chains ($N>15$).
> >
> > **[R1]** Holzleitner, Markus, et al. "Convergence proof for actor-critic methods applied to ppo and rudder." Transactions on Large-Scale Data-and Knowledge-Centered Systems, 2021. 105-130.
> >
> > **[R2]** Konda, Vijay, and John Tsitsiklis. "Actor-critic algorithms." Advances in neural information processing systems (1999).
> >
> > **[R3]** Engstrom, Logan, et al. "Implementation matters in deep policy gradients: A case study on ppo and trpo." arXiv preprint arXiv:2005.12729 (2020).
> >
> > **[R4]** Andrychowicz, Marcin, et al. "What matters for on-policy deep actor-critic methods? a large-scale study." International conference on learning representations, 2020.
> >
> > **[R5]** Hottung, André, Bhanu Bhandari, and Kevin Tierney. "Learning a latent search space for routing problems using variational autoencoders." International conference on learning representations, 2020.
> >
> > **[R6]** Chandak, Yash, et al. "Learning action representations for reinforcement learning." International conference on machine learning, 2019.

---

> > > ### Comment · Reviewer_MBaj · 2023-11-23
> > > **I acknowledge that I have read the authors' comments**
> > >
> > > Thanks for the detailed response and for making the modifications. My concerns and questions have been addressed.

---

> > > > ### Author Response · Authors · 2023-11-23
> > > > **Thanks to the Reviewer MBaj**
> > > >
> > > > Thank you for your positive feedback. We sincerely appreciate your valuable suggestions and time.

---

### Author Response · Authors · 2023-11-22
**General Response to All Reviewers**

Dear Reviewers,

We sincerely thank all the reviewers (MBaj,Sjm2,554U,rNSL) for their valuable feedback. We are glad that the reviewers appreciated the significance of our problem (MBaj,Sjm2), the interest and novelty of our proposed framework (MBaj), the comprehensiveness of our experiments (MBaj,554U), and the quality of our paper's writing (554U,rNSL).

We have made every effort to faithfully address the comments in the responses. As suggested by the reviewers, we have made the following modifications and additions to our manuscript.

* We add additional experimental results of our method with AFM- and ESMFold-produced dimers. Results are shown in the revised Table 2. (MBaj,rNSL)
* We add more descriptions about terminology (i.e., docking, chain, dimer, complex) to the Appendix A. (Sjm2)
* We add the visualized results (three cases) of the assembly process predicted by our method to Appendix C.3. (MBaj)
* We add the convergence episode values over different chain numbers to the Appendix C.4 (Table 7 and Figure 8). (MBaj)
* We add the explanation for the balance between exploration and exploitation to the Appendix E. (Sjm2)
* We add the analysis of total running time to the Appendix F. (rNSL)
* We revise the statements for the term $p_{data}$ and the "AGG" notation in Section 3.5. (Sjm2)

We have incorporated the suggested modifications in the revised manuscript version, which are highlighted in blue. As the deadline for discussion is fast approaching, we would greatly appreciate it if you could allocate some time to review our responses.

Thanks for all reviewers' time again.

---

### Meta-Review · Area_Chair_uSk7 · 2023-12-11

**Metareview:**

The paper introduces the Generative Adversarial Policy Network (GAPN), a reinforcement-based method for modelling large protein complexes through sequential assembly of individual protein chains. GAPN uses adversarial reward functions, incorporating knowledge from various-sized complexes for better generalization, and explores the combinatorial search space to identify optimal assembly actions. The empirical results show that GAPN surpasses existing state-of-the-art methods in performance. The reviewers found that the paper was well-motivated and well-organized, introduced a novel integration of reinforcement learning, graph neural networks, and adversarial training, and demonstrated significant accuracy and efficiency improvements. However, they also pointed out several weaknesses of the paper, including the reliance on precomputed dimer structures, which may limit real-world applicability, and that the protein docking process is oversimplified, potentially leading to error accumulation. The paper could benefit from more detailed explanations of key concepts and methodologies.

**Justification For Why Not Higher Score:**

The reviews have pointed out several limitations in the paper, including the reliance on precomputed dimer structures and the limitations in real-world applicability, and consequently recommend borderline acceptance.

**Justification For Why Not Lower Score:**

Despite its limitations, the proposed method is novel, and the paper demonstrates improvements in accuracy and efficiency, which are significant contributions to the field.

---

### Decision · Program_Chairs · 2024-01-16

Accept (poster)